# Never Too Late to Train: The Effects of Pelvic Floor Muscle Training on the Shape of the Levator Hiatus in Incontinent Older Women

**DOI:** 10.3390/ijerph191711078

**Published:** 2022-09-04

**Authors:** Licia P. Cacciari, Mélanie Morin, Marie-Hélène Mayrand, Chantale Dumoulin

**Affiliations:** 1School of Rehabilitation, Faculty of Medicine, Université de Montréal and Research Center of the Institut Universitaire de Gériatrie de Montréal, Montreal, QC H3W 1W5, Canada; 2School of Rehabilitation, Faculty of Medicine and Health Sciences, Université de Sherbrooke, Research Center of the Centre Hospitalier de l’Université de Sherbrooke, Sherbrooke, QC J1H 5N4, Canada; 3Departments of Obstetrics and Gynecology and Social and Preventive Medicine, Université de Montréal, Research Center of the Centre Hospitalier de l’Université de Montréal, Montreal, QC H3C 3J7, Canada

**Keywords:** pelvic floor physiotherapy, urinary incontinence, conservative management, older women, pelvic floor ultrasound, pelvic floor morphometry

## Abstract

Pelvic floor muscle (PFM) training is the first-line treatment for women of all ages with urinary incontinence (UI), but evidence supporting its effects on the functional anatomy of the pelvic floor is scarce in older women. We aimed to evaluate the long-term (one-year) effects of PFM training on the shape of the levator hiatus (LH) in older women with UI and its association with PFM force, incontinence severity, and potential effect modifiers (age, UI severity, BMI, and UI type). This is a secondary analysis of the GROUP study, a non-inferiority RCT assessing the effects of a structured and progressive 12-week PFM training program to treat UI in older women. Data were available from 264/308 participants at the one-year follow-up. PFM training resulted in reduced LH size toward a more “circular” shape, which was consistently associated with greater PFM force and reduced UI severity. Further, no significant interactions were found between LH shape changes and any of the potential effect modifiers, suggesting that women will potentially benefit from PFM training, regardless of age, UI severity, BMI, and UI type (stress or mixed), with changes that can be observed in the functional anatomy of the pelvic floor and sustained in the long-term.

## 1. Introduction

Pelvic floor muscle (PFM) training is the recommended first-line treatment for women of all ages with urinary incontinence (UI) [1,2]. The benefits of PFM training have been shown across age cohorts [3,4], but in older women evidence is lacking for its effects on the functional anatomy of the pelvic floor.

We previously demonstrated that older women will attend and adhere to PFM training, and gain clinically important benefits, including a 70–74% reduction in UI episodes [4]. Additionally, improvements were found in overall PFM function (as assessed by dynamometry), but these were not always evident in the functional anatomy of the pelvic floor (as assessed by transperineal ultrasound) [5]. More specifically, training-related changes were found in the levator hiatus (LH) diameter and area while women were coughing, but not at rest or during a PFM contraction as expected. A possible explanation could be a higher variability within morphometric parameters common to this population when compared to reports in asymptomatic younger women [6,7].

Aging and menopause have been linked to continuous atrophy and loss of PFM force [8,9]. Recently, two pilot studies comparing older to younger nulliparous women found age-related changes in the LH form or shape [10,11]. More specifically, LH shape was defined as the ratio between the LH sagittal and coronal diameters, and older women presented a more oval LH shape, which was also associated with more severe UI symptoms [11].

We hypothesized that the shape of the LH, as opposed to the usual single LH measures, was a potentially useful strategy to allow a more comprehensive measure of pelvic floor function in older women, and to normalize individual LH measures as a mean to reduce variability. By using a ratio, we expect to reduce random errors related to the assessment condition (transducer pressure/position) [12] that could mask discreet effects related to the intervention. 

In this context, we sought to further understand the long-term effects of PFM training on the functional anatomy of the pelvic floor, more specifically concerning the shape of the LH as assessed by transperineal ultrasonography. We further explored the relationship between LH shape and PFM force, UI severity and potential effect modifiers. 

## 2. Materials and Methods

This is a secondary analysis of a two-armed RCT designed to assess the noninferiority of group-based compared to individual PFM training to treat UI in older women (ClinicalTrials.gov NCT02039830). Results of the primary analysis showed improvements in UI-related signs, symptoms [4], and PFM function [5].

The study was conducted at two centers (Université de Montréal and Université de Sherbrooke, Canada) from July 2012 to June 2018. The population included women aged 60 and over, with symptoms of stress or mixed UI (at least three episodes of involuntary urine loss per week during the preceding three months). Stress/mixed UI was confirmed using the validated questionnaire for incontinence diagnosis. Trial exclusion criteria included a body mass index (BMI) of 35 or greater, reduced mobility, chronic constipation, and significant pelvic organ prolapse (pelvic organ prolapse quantification system > stage 2) [4]. In this analysis, participants were also excluded if ultrasound datasets were missing or unusable (e.g., unwillingness to do the exam, incomplete imaging of the hiatus, or poor imaging conditions). All women signed an informed consent form prior to inclusion, and the study was approved by the research ethics boards at both research centers of the Institut Universitaire de Gériatrie de Montréal and the Centre Hospitalier Universitaire de Sherbrooke. The study sample size was based on power calculations for the primary outcome in the parent study, which was the percentage reduction in UI episodes [13].

The intervention comprised a structured and progressive 12-week PFM training program delivered by experienced physiotherapists. This training protocol was previously deemed acceptable, suitable, and efficient to treat older women with UI [14]. All women attended an individual session with a physiotherapist to learn how to effectively contract their PFMs using vaginal palpation. This was followed by 12 weekly one-hour sessions, either individually (one-on-one) or in groups of six to eight women. In both cases, each session included a 15-min educational period and a 45-min exercise component including virtual reality rehabilitation. The exercise protocol was structured with a gradual increase in difficulty in terms of exercise duration, number of repetitions, and position (including lying down, sitting, four-point kneeling, and standing). The virtual reality rehabilitation consisted of a dance game (www.stepmania.com, accessed on 3 September 2022) adapted to include PFM training in a functional, fun, and challenging format. Additionally, women in both study arms were expected to perform exercises at home, five days per week, during the 12-week PFM training program, and subsequently three days per week for nine months. Session attendance was monitored by the treating physiotherapists. Adherence to the home exercise program was assessed using the participants’ exercise diaries during treatment and by telephone follow-ups at six, nine, and twelve months post-intervention. Only participants who attended 10 or more of the 12 sessions were included in the present analysis. Further details are provided in the trial protocol [13] and main trial [4].

Outcomes were acquired at baseline and follow-up (one-year post randomization) and included: (1) LH shape, as assessed by transperineal ultrasound (US), (2) PFM force, as assessed by intravaginal dynamometry, and (3) UI symptom severity, based on the International Consultation on Incontinence Questionnaire-Urinary Incontinence Short Form (ICIQ-UI SF).

LH shape was obtained through transperineal US volumes using either a Siemens Acuson Antares system with a 3–5 MHz curvilinear three-dimensional/four-dimensional probe or a GE Voluson Expert system with a 2–6 MHz curvilinear three-dimensional/four-dimensional probe, depending on equipment availability at each study center. Women were asked to empty their bladder, and then the examination was performed in the supine position with hips and knees flexed. The US probe was placed on the perineum in a midsagittal plane. The volumes assessed included the posteroinferior margin of the symphysis pubis up to the back sling of the puborectalis muscle. Data acquisition was performed: (1) at rest, (2) during a maximal PFM contraction, and (3) during a strong cough. Each condition was repeated twice and followed by a 10-s relaxation period. Volume datasets were analyzed offline (4D View, Version 10.2, GE Healthcare, Zipf, Austria or Syngo FourSight ViewTool, version 3.1, Siemens Canada Ltd., Mississauga, ON, Canada) by an independent assessor, who was blinded to the evaluation time point. The best trial of each condition was considered for analysis. When possible, partial data analysis was performed (if data was not available for all assessment conditions). Hiatal diameters were measured at the plane of minimal hiatal dimension as defined in the mid-sagittal plane, which was observed as the minimal distance between the hyperechogenic posterior aspect of the symphysis pubis and the hyperechogenic anterior border of the levator ani muscle, just posterior to the anorectal angle (Figure 1) [15]. LH shape was calculated as the ratio between the LH sagittal and coronal diameters both measured in millimeters [11] (see Figure 1). Inter-rater repeatability of both LH sagittal and coronal diameters was previously tested in a similar population and found to be “good” to “very good” (intraclass correlation coefficients = 0.63–0.90) [16].

PFM force was assessed using the Montreal dynamometer, which has been extensively described in prior studies [17]. Briefly, this is a custom-built instrumented speculum, which includes two parallel aluminum branches that are both fixed to a base. The lower branch is adjustable to different apertures and includes two strain gauges are mounted in a differential arrangement to capture PFM contractions independent of the precise site of application. Following a previously validated protocol [17], PFM force was measured in women in the supine position, with hips and knees flexed at rest, during a maximal PFM contraction and a triple cough [5]. For the maximal contraction and triple cough tests, force magnitude was subtracted from a baseline mean value obtained before each trial. UI symptom severity was based on the ICIQ-UI SF total score (from 0 to 21, greater values indicate increased severity) [18], and a validated and highly recommended questionnaire [1] that considers UI frequency, severity, and impact on quality of life.

Once the non-inferiority hypothesis was confirmed in the main paper (group-based was non-inferior to individual PFM training) [4] and both study arms were comparable at all time points in terms of PFM morphometry and function [5], data from both study arms were combined for this analysis. To assess the long-term effects of PFM training on LH shape, we used mixed-effects models to minimize the effect of missing data. The homogeneity assumption was assessed using a residual vs. fitted values plot, and the assumption of normality of residuals was assessed using a Q–Q plot. The Pearson’s correlation coefficient (r) was used to assess the correlation between LH shape and PFM force at the baseline and follow-up. Linear regression assessed whether the LH shape was associated with UI severity at the follow-up. Subgroup analyses and interaction terms or linear regression were used to assess the interaction between the changes in LH shape during contraction and potential effect modifiers (age, UI severity, BMI, and UI type-established in the trial protocol) [13]. Two-sided *p*-values < 0.05 were considered statistically significant. Analyses were completed using R Version 4.0.2, R Foundation for statistical computing, Vienna, Austria and SPSS Statistics for Windows, Version 24.0, IBM Corp, Amonk, NY, USA.

## 3. Results

A total of 362 participants were enrolled in the main trial, and US volumes were available from 308 (85%) and 264 (73%) of the participants at baseline and follow-up, including at least one assessment condition (rest, contraction, or cough). The precise sample size available for each comparison is specified in each table. From the included participants (*n* = 308), mean age was 68 years old (±5.8, range from 59 to 89), they were on average overweight with a mean BMI of 27 (±4.5, from 16 to 40), and the median parity was two (range, 0–8). Two hundred and fifty-three participants (82%) had symptoms of mixed UI, and 55 (18%) had symptoms of stress UI. The mean duration of UI symptoms was 9.3 (±9.5) years and the mean leakage episodes per week was 14.5 (±14.5). 

Table 1 summarizes the effect of PFM training on LH shape at the one-year follow-up. Of interest, PFM training resulted in reduced LH size towards a more “circular” shape during both PFM contraction and cough. In both conditions, the change effect was small but consistent (narrow confidence intervals), suggesting a sustained impact of rehabilitation on the functional anatomy of the pelvic floor. No changes were observed when participants were assessed at rest.

Table 2 summarizes the correlation between LH shape and PFM force assessed using dynamometry. Overall, negative weak correlations were found across the three assessment conditions (at rest and during contraction and cough) and time points (baseline and one-year follow-up). Of interest, a smaller and more circular shaped LH was related to greater PFM force (r = 0.01 to −0.33); greater change toward a more circular shape was related to greater force gains (r = 0.08 to −0.16); and a larger LH at baseline (more oval shaped) was linked to the greatest change toward a more circular shape at follow-up (r = −0.22 to −0.34). 

Consistently, an association was found between LH shape during contraction and UI severity at follow-up (average increase of 2.7 points in incontinence severity [ICIQ-UI SF] per unit increase in LH shape ratio on contraction, 95% confidence interval 0.3 to 5.0, *p* = 0.03), where women with a smaller and more circular shaped LH presented lower UI symptoms. Finally, no statistically significant interactions were found between the changes in LH shape during contraction (follow-up—baseline) and any potential effect modifiers identified a priori, including age, UI severity, BMI, and UI type (Table 3).

**Table 3 ijerph-19-11078-t003:** Association between the changes in shape of the levator hiatus during contraction and outcomes of interest.

Outcomes	*n* (Baseline/One Year)	Baseline UI Severity	One Year UI Severity	Interaction
Binary outcomes				
Age, years				
<70	211/180	1.21 (0.21)	1.18 (0.21)	0.59
>70	84/70	1.26 (0.21)	1.25 (0.23)	
UI severity ^a^				
Moderate	148/128	1.22 (0.22)	1.19 (0.22)	0.74
Severe	146/121	1.23 (0.21)	1.21 (0.22)	
BMI, kg/m^2 b^				
Normal	101/85	1.19 (0.20)	1.16 (0.21)	0.92
High	194/165	1.24 (0.21)	1.22 (0.23)	
UI type				
Stress UI	53/47	1.20 (0.20)	1.17 (0.24)	0.48
Mixed UI	242/203	1.23 (0.21)	1.21 (0.22)	
Continuous outcomes				
Age	295/250			0.15
Severity	295/250			0.42
BMI	294/249			0.32
Age × Severity	295/250			0.29
Age × BMI	294/249			0.59
BMI × Severity	295/250			0.92

Data are given as means (SD). UI, urinary incontinence; BMI, body-mass index, calculated as weight in kilograms divided by height in meters squared. ^a^ UI severity was based on International Consultation on Incontinence Questionnaire-Urinary Incontinence Short Form (ICIQ-UI SF); scores higher than 12/21 were considered severe [19]. ^b^ A body mass index over 25 kg/m^2^ was considered high [20].

## 4. Discussion

To our knowledge, this is the first study to assess the long-term effects of PFM training on the shape of the LH in incontinent older women, as opposed to individual measures of LH area or dimension. PFM training resulted in a reduced LH size to a more “circular” shape. Overall, differences were small (small effect) but consistent across assessment conditions (contraction and cough). Accordingly, a more circular LH was related to improved PFM strength and reduced UI symptoms. Women with more circular-shaped LHs produced higher PFM forces, and those with a greater reduction in LH size toward a more circular shape presented a greater improvement in PFM force. 

The fact that women with larger LH (more oval shaped) exhibited the greatest change toward a more circular shape, together with the absence of interactions between LH shape change and any potential effect modifiers (age, UI severity, BMI, and UI type), strengthen the evidence that one cannot be too old, too incontinent, or too overweight to benefit from PFM training. Furthermore, training effects were not related to UI type, suggesting that women with both stress and mixed UI will benefit from the intervention. We acknowledge, however, that the exclusion criteria (reduced mobility, BMI > 35) and intensive intervention could limit the generalizability of the results to frail or morbidly obese women, or to those who, for some reason, are unable to participate or adhere to the proposed training protocol.

Nevertheless, our results are consistent with early evidence reporting morphometrical changes in pelvic floor structures following PFM training in younger women, including smaller LH area at rest [21] and increased displacement of the bladder [22,23]. In a prior publication, these differences did not reach statistical significance in older women [5], possibly due to a smaller effect and higher variability common to this population. By assessing the LH shape as a ratio between the LH sagittal and coronal diameters, we were able to highlight the changes related to treatment. Furthermore, this seems to be a simple approach to quantify the efficiency of the PFM, as opposed to the assessment of individual measures of the LH area or diameter, which may not be as representative of overall pelvic floor function.

Although established as 1st line treatment for all women with UI, PFM training is still not routinely available to many incontinent women worldwide [24]. More specifically, those with presumed determinants of treatment failure (mixed UI, older age, or severe UI symptoms) are often directly directed to either drug treatment (mixed UI) [25] or surgery (older age or severe UI) [26,27] despite the known adverse effects [28]. Knowing that PFM training effects on LH shape were sustained and consistent across potential effect modifiers only reinforces the recommendation that PFM training should be made more available and offered routinely for all women with UI. Future studies are needed to further explore determinants to long-term LH shape change across different age groups.

## 5. Conclusions

PFM training resulted in sustained changes in LH shape in older women. Although changes were small, they were consistent, not impacted by potential effect modifiers and related to higher PFM force and lower UI symptoms. From a clinical perspective, our findings strengthen the recommendation that all women will potentially benefit from PFM training, regardless of age, UI severity, and UI type (stress or mixed), with changes that can be observed in the functional anatomy of the pelvic floor.

## Figures and Tables

**Figure 1 ijerph-19-11078-f001:**
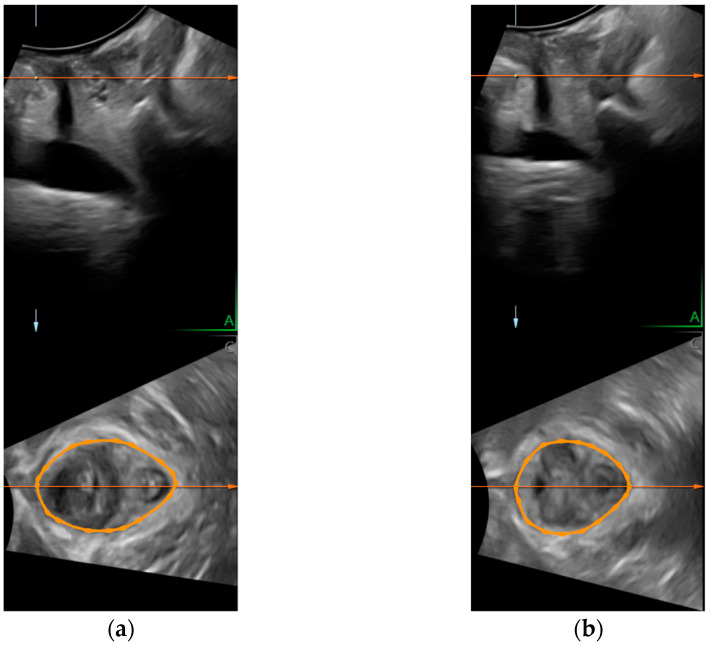
Example of a transperineal ultrasound image in the plane of minimal hiatal dimensions showing the levator hiatus (orange circle), which has changed into a more circular shape following pelvic floor muscle (PFM) training. Images were taken during a PFM contraction: (**a**) at baseline; (**b**) at the one-year follow-up.

**Table 1 ijerph-19-11078-t001:** Levator hiatus shape following pelvic floor muscle training for urinary incontinence at baseline and after one year.

Assessment Condition	*n* Baseline/ One-Year	Baseline Hiatus Shape	One-Year Hiatus Shape	Mean Difference (95% CI)	*p*	Cohen’s d (95% CI)
Rest	290/244	1.39 (0.24)	1.39 (0.24)	−0.01 (−0.03 to 0.02)	0.630	0.04 (−0.09 to 0.17)
Contraction *	295/250	1.22 (0.21)	1.20 (0.22)	−0.02 (−0.04 to −0.01)	0.005	0.18 (0.06 to 0.31)
Cough *	285/244	1.31 (0.21)	1.29 (0.22)	−0.02 (−0.04 to 0.00)	0.033	0.14 (0.01 to 0.27)

Data are given as mean (SD). * *p* < 0.05 (2-tailed). Hiatus shape calculated as the ratio between the LH sagittal and coronal diameters.

**Table 2 ijerph-19-11078-t002:** Correlation between levator hiatus shape and pelvic floor muscle force at rest, and during contraction and cough.

Parameter	*n*	Pearson’s Correlation
Rest		
LH shape at baseline vs. PFM force at baseline *	233	−0.13
LH shape at one year vs. PFM force at one year **	232	−0.20
LH shape change vs. PFM force change	205	0.08
LH shape baseline vs. LH shape change **	230	−0.34
PFM contraction		
LH shape at baseline vs. PFM force at baseline **	274	−0.32
LH shape at one year vs. PFM force at one year **	233	−0.33
LH shape change vs. PFM force change *	214	−0.16
LH shape at baseline vs. LH shape change **	241	−0.22
Cough		
LH shape at baseline vs. PFM force at baseline	260	0.01
LH shape at one year vs. PFM force at one year **	223	−0.25
LH shape change vs. PFM force change *	191	−0.16
LH shape at baseline vs. LH shape change **	223	−0.32

PFM, pelvic floor muscle; LH, levator hiatus; * Correlation is significant at the 0.05 level (2-tailed). ** Correlation is significant at the 0.01 level (2-tailed).

## Data Availability

The data presented in this study are available on request from the corresponding author.

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
