# Peer review of "Never Too Late to Train: The Effects of Pelvic Floor Muscle Training on the Shape of the Levator Hiatus in Incontinent Older Women"

_ijerph, 2022, doi:10.3390/ijerph191711078_

Round 1
Reviewer 1 Report
The work is interesting. PFM training could be described more precisely. In the future, I suggest combining PFM with other types of interventions, such as the sonofeedback method.
Reviewer 2 Report
The authors present a retrospective secondary analysis of a clinical trial on the effects of a 12-week pelvic floor training program to treat urinary incontinence in older women. They found that pelvic floor training reduced levator hiatus size toward a more circular shape, which was associated with greater pelvic floor muscle force and reduced urinary incontinence severity. While prior studies have demonstrated an effect of pelvic floor training on levator hiatus size in younger women, this is the first study to demonstrate this in older women. The study is well conducted, and the article is well written. The manuscript might be improved by considering the following comments.
Table 1. Please include units in the table or the caption. The table should be able to stand independently and make sense to someone who has not read the article. I would also replace “baseline” and “one-year” with more descriptive terms, like “baseline hiatus shape” and “one-year hiatus shape.”
Results: Paragraph 4: Sentence 1: Please include units. It is not clear what is meant by “estimate 2.7.”
Table 3. The table title says “This is a table. Tables should be placed in the main text near to the first time they are cited.” Did the authors mean to use this as the title for their table? I would also replace “baseline” and “one-year” with more descriptive terms, like “baseline UI severity” and “one-year UI severity.”
Discussion. While this study adds to our scientific understanding of pelvic floor training, it does not have a major clinical impact. Elderly women are already routinely referred for pelvic floor training as the clinical results are well documented. As this is more of a scientific study than a practice changing study, the authors should elaborate on the scientific implications of these findings. What are the next research steps? Can any novel therapeutic or diagnostic advances potentially stem from this line of research in the future, or is the goal simply to continue referring elderly women with urinary incontinence to pelvic floor training?
